# A Perspective on Body Size and Abundance Relationships across Ecological Communities

**DOI:** 10.3390/biology9030042

**Published:** 2020-02-26

**Authors:** Vojsava Gjoni, Douglas Stewart Glazier

**Affiliations:** 1Department of Biological and Environmental Sciences and Technologies, University of Salento, Via Monteroni, Ecotekne, 73100 Lecce, Italy; 2Department of Biology, Juniata College, Huntingdon, PA 16652, USA; glazier@juniata.edu

**Keywords:** abundance, body size, ecological communities, scaling, size-abundance relationship

## Abstract

Recently, several studies have reported relationships between the abundance of organisms in an ecological community and their mean body size (called cross-community scaling relationships: CCSRs) that can be described by simple power functions. A primary focus of these studies has been on the scaling exponent (slope) and whether it approximates −3/4, as predicted by Damuth’s rule and the metabolic theory in ecology. However, some CCSR studies have reported scaling exponents significantly different from the theoretical value of −3/4. Why this variation occurs is still largely unknown. The purpose of our commentary is to show the value of examining both the slopes and elevations of CCSRs and how various ecological factors may affect them. As a heuristic exercise, we reanalyzed three published data sets based on phytoplankton, rodent, and macroinvertebrate assemblages that we subdivided according to three distinctly different ecological factors (i.e., climate zone, season, and trophic level). Our analyses reveal significant variation in either or both the CCSR slopes and elevations for marine phytoplankton communities across climate zones, a desert rodent community across seasons, and saltwater lagoon macroinvertebrate communities across trophic levels. We conclude that achieving a comprehensive understanding of abundance-size relationships at the community level will require consideration of both slopes and elevations of these relationships and their possible variation in different ecological contexts.

## 1. Introduction

Since the 1930s, researchers observed that average plant size was inversely related to population density. This density-dependent effect is observed by comparing several populations of the same species. The common occurrence of this relationship, which often shows a log-log scaling exponent of −3/2, has led to it being considered a rule or law, often called the self-thinning rule (STR), the −3/2 power rule, or Yoda’s law [1]. The STR is so general that it has even been applied to animals, though the scaling exponent may take on other values, such as −4/3. The generality of the STR has thus spurred many investigators to explain it in both plants [2,3,4] and animals (e.g., insects [5,6,7], marine invertebrates [8,9,10,11,12,13] and fish [14,15,16,17,18,19]). 

Recently, the STR has also been extended to include comparisons of ecological communities of multiple species, and not just conspecific populations. Remarkably, power functions have also been successfully applied at the community level, thus starting a new wave of analyses of cross-community scaling relationships (CCSR). CCSRs describe negative relationships between the total number or density of organisms in an assemblage or community of species and their average body size [20]. These relationships are often so regular that they can be described by the power function: *N* = *kB^b^*,(1)
where *N* is population density, *k* is a normalization constant, *B* is individual body mass, and *b* is the scaling exponent. According to Damuth’s Rule, which is based on comparisons of species populations, the scaling exponent should be −3/4 [21,22,23,24]. The ‘metabolic theory of ecology’ makes a similar prediction, by assuming that the scaling exponent for population density should be the inverse of that based on the 3/4-power law for metabolic rate [25]. This assumption derives from the Energetic Equivalence Rule (EER), which posits that species populations use approximately the same amount of energy regardless of body size (calculated as energetic demand per individual, scaled to body mass according to the 3/4-power law, times the number of individuals in a population [23,24]).

However, when ecological densities are compared among community assemblages of closely related species, the slope of the CCSR may deviate from the theoretical value of −3/4. These deviations may occur if the EER is not obeyed. For example, a less steep slope may occur if large species acquire more energy than smaller species. In contrast, a steeper slope may occur if small species acquire more energy than larger species [21,22,23]. These deviations from that expected by the EER or MTE show that the amount of energy used is not the same for all species, which may result from the effects of various biotic and abiotic environmental factors. 

To date, several studies of CCSRs have been carried out on diverse assemblages of plants [26], phytoplankton [27], bacteria, algae and protozoa [20,28], fish, amphibians and macroinvertebrates [29,30,31,32], birds [33], and rodents [34]. However, the focus of these studies has been chiefly on the scaling exponent and whether it matches the theoretically predicted value of −3/4. In this commentary, we advocate expanding the focus of studies of CCSRs to include explorations of the biological meaning of both their slopes and elevations, especially in relation to various intrinsic (biological) and extrinsic (ecological) factors. We support this view by re-analyzing selected data sets from previously published papers to show that either the slopes or elevations (or both) of CCSRs vary considerably in relation to three distinctly different ecological factors: climate zone, season and trophic level. The results of our heuristic exercise add to the growing literature showing that diverse kinds of biological and ecological scaling relationships do not necessarily follow simple universal laws, but vary substantially in relation to various biological and ecological contexts (e.g., [35,36,37,38,39,40]).

## 2. Case Studies

### 2.1. Across Climate Zones (Biogeographic Regions)

The scaling slope for the CCSR of 635 marine phytoplankton community assemblages in the Northern Hemisphere of the Atlantic Ocean was found to be −0.78 (95% confidence interval = −0.74 to −0.81, based on reduced major axis (Model II) regression; Figure 1A), which does not differ significantly from the theoretical value of −3/4 [27]. This analysis and other phytoplankton studies [4,41] support the generality of the −3/4-power rule of Damuth and the MTE (but note that a least squares (Model I) regression (LSR) analysis of the data of [27] yields an exponent (−0.68: Table 1) significantly less than −3/4). However, plankton communities consisting of both phyto- and zooplankton show CCSR scaling exponents of −1, apparently resulting from a dominance of small-sized species [42]. Here we show that the CCSR scaling exponent may vary not only with plankton species composition, but also with climate zone or biogeographic region.

We divided the dataset of [27] according to four climate zones. Our LSR analyses show that the CCSR exponents and intercepts for cell density versus mean cell size (carbon content) vary significantly across the four climate zones (Figure 1B). In particular, although the slopes are not significantly different from −3/4 in the assemblages occupying the southern climate zones (Gulf Stream and Northwest Atlantic Shelves), they are significantly lower than −3/4 in the assemblages occupying the northern climate zones (Atlantic Arctic and Boreal Polar) (see Table 1, Figure 1B).

Furthermore, size-specific cell densities (and thus overall scaling elevations) tend to be higher in the northern vs. southern climate zones (ANCOVA analysis comparing 95% confidence interval; Table 2). This could be explained hypothetically as the result of two major influences: larger cells are favored in colder more northerly climate zones (following the temperature size rule, as the data seem to show) and lower temperatures cause lower metabolic (nutrient) demand per cell, thus enabling higher total cell densities [43,44,45]. 

This hypothesis may also help explain why the scaling slopes of the CCSRs are lower in the northern versus southern climate zones. This difference may arise because although small-celled species show similar densities in all climate zones, larger-celled species show significantly higher densities in northern vs. southern climate zones. This may be because much fewer small-celled species occur in the northern climate zones, thus freeing up resources (nutrients) for larger-celled species that can then build up higher densities. However, many small-celled species occur in the southern climate zones, and their competitive exploitation of shared nutrients limits the densities of larger-celled species to lower levels. In short, a shift in competitive advantage from small to large cells with increasing latitude and decreasing temperature may cause changes in both the scaling slopes and elevations of the CCSRs observed.

### 2.2. Across Seasons

The CCSR of a desert rodent community also shows an exponent not significantly different from the theoretical value of −3/4 [34]. LSR analysis of overall abundance with mean body size at the same sampling site in Portal Arizona over a 25-year period (1987 to 2002), showed a CCSR exponent of −0.57 (*n* = 25, 95% confidence interval: −0.97 to −0.18). The CCSR was constructed from community assemblages at different time periods (rather than different spatial locations, as typically done). Each data point was based on monthly sampling during a specific year, which was performed three or four times. However, the data set includes only the species that occurred during a 6-month period in at least five years during the 25 years period of the data set.

We obtained an updated data set covering 41 years of sampling, kindly provided by Ethan White, and divided it according to season. Our LSR analysis of this dataset, including all species sampled, yielded a CCSR scaling exponent similar to that found over the 25-year period analyzed by [34], and not significantly different from the theoretical value of −3/4 (Table 3, Figure 2A). Moreover, the CCSR scaling exponents for each of the four seasons did not differ significantly from each other (Table 3, Figure 2B), nor with −3/4. However, the CCSR elevation was significantly higher during the spring than during the winter (ANCOVA analysis comparing 95% confidence interval; Table 4).

Rodents may have reached their highest density during spring, because this is when reproduction and the appearance of juveniles tend to peak for most species [46,47,48]. However, during the winter, density is relatively low because breeding is low, thus causing a net loss of individuals by mortality. The relative heights of the elevations of these relationships are consistent with the proposed hypothesis. As shown in Figure 2B, rodent densities tended to be lowest during winter, rising to the highest level during spring, and then declining again through summer and autumn. Therefore, our analysis suggests that although the energy flow of the desert rodent community has apparently not changed over the 25-year period studied by [34], it has changed seasonally during each year (as indicated by seasonal changes in the scaling elevation for population density). 

### 2.3. Across Trophic Levels

Although some CCSR scaling exponents reported in the literature so far are not significantly different from the theoretical value of −3/4 ([27,32,33,34]; see also Section 3), a recent analysis of 158 aquatic macroinvertebrate community assemblages has revealed an exponent significantly lower than −3/4 (−0.27, 95% confidence interval = −0.411 to −0.131) [29]. This data set included spring and summer samples from saltwater lagoons in the biogeographic regions of the Eastern Mediterranean Sea and the Black Sea. It suggests that large macroinvertebrate species seem to be acquiring more energy than smaller species. 

We reanalyzed the data set of [29] by averaging the data for spring and autumn. Again, the CCSR scaling exponent (−0.43) is not significantly different from the value obtained by [29], and is still significantly lower than −3/4 (Table 5, Figure 3A). Furthermore, after separating the data by trophic level, the CCSR scaling exponent is significantly lower than −3/4 for predator and prey species analyzed separately (−0.45 and −0.39, respectively: see Table 5, Figure 3B). However, the elevation of the scaling relationship is significantly higher for prey than predators (ANCOVA analysis comparing 95% confidence intervals; Table 6). 

The higher CCSR elevation for prey vs. predators may be explained in terms of both general theory and observations specific to saltwater lagoons. Following the second law of thermodynamics, as energy is transformed from lower to higher trophic levels in a food web, much energy is lost as heat [49,50]. Therefore, since prey tend to have more energy available to them than predators that feed upon them, they can also sustain higher population densities at equivalent body sizes. In addition, macroinvertebrate prey in shallow, highly productive, saltwater lagoons are mainly detritus and suspension feeders. The energy available to them from abundant, easily acquired detritus and fine organic matter in the water column and at the bottom of lagoons is much more abundant than that available to predators that feed chiefly on animal tissue, which is more difficult to acquire [51,52,53]. Our results are consistent with previous findings showing that prey tend to have higher population abundances than that of predators, even if predators affect the abundance of prey [54,55,56,57,58].

## 3. Discussion

Most previous studies of abundance-size relationships across ecological communities have focused on the scaling exponent, and whether it is similar to the −3/4 values predicted by Damuth’s rule and the MTE [21,22,23,24,25]. Although three major studies have yielded CCSR exponents not significantly different from −3/4, three other studies have reported exponents significantly higher or lower than −3/4 (Table 7). Why this is so is still little understood. Some of this variation may relate to taxonomic and/or environmental differences.

In our commentary, we suggest that better understanding of CCSRs may be achieved by examining both the slopes and elevations of these relationships, and how they are affected by various ecological factors. To support this point, we reanalyzed three data sets published in the literature. Instead of examining these data sets as a whole, as done by the original authors, we divided them using various ecological factors, including climate zone, season and trophic level. In all of our case studies, we found significant variation in CCSR slopes and/or elevations (intercepts) among our ecologically classified subsamples. 

First, we found significant differences in slopes and elevations between CCSRs of marine phytoplankton communities from northern cold climate zones versus southern warmer climate zones that could be explained in terms of plausible temperature effects on the cell size, metabolic rate and density of phytoplankton (Section 2.1). Second, we discovered significant seasonal differences in the elevations of CCSRs based on temporal samples of a desert rodent community that could be explained as a result of seasonal differences in offspring production (Section 2.2). Third, we found significant differences in the elevations of CCSRs for prey versus predators of saltwater lagoon macroinvertebrate communities that could be explained by lower availability of energy at higher trophic levels (Section 2.3). 

All of these case studies reveal the value of subdividing ecologically heterogeneous data sets into more homogeneous ecological categories. By doing so, significant differences in CCSRs may be found that can help elucidate the mechanisms underlying them. Our analyses also reveal the importance of exploring the biological meaning of both the slopes and elevations of CCSRs, as recommended in general for allometric scaling analyses (e.g., [35,36,59,60]).

## 4. Conclusions

Although abundance−size relationships have received much attention by ecologists at the population level, little is known about these relationships at the community level. We hope that our analyses will serve as heuristic examples that will motivate other researchers to explore ecological and taxonomic effects on both the slopes and elevations of CCSRs. Although universal laws are useful for testing theoretical expectations on major ecological issues, various contingent factors may cause many kinds of biological and ecological scaling relationships to deviate from universal laws [34,35,36,37,38,39,40]. We suggest that the search for and understanding of regular patterns in nature benefit from not only employing general theories based on single, hypothetically universal, deterministic mechanisms, but also an awareness that multiple contingent mechanisms that vary with biological and ecological context may underlie the diversity that we often see. 

## Figures and Tables

**Figure 1 biology-09-00042-f001:**
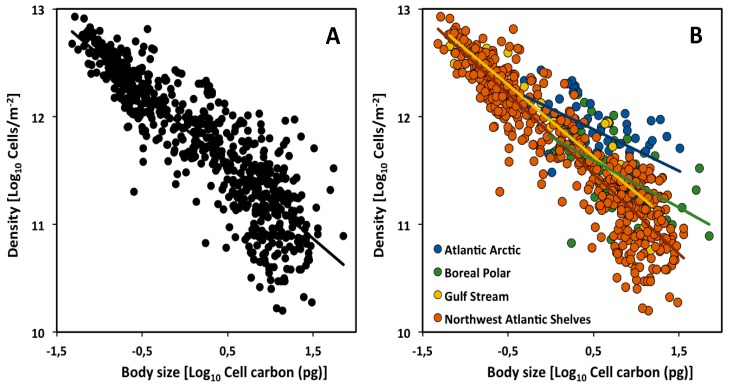
Scaling of cell density versus cell size (carbon mass) of phytoplankton communities in the Atlantic Ocean (data from [27]): (**A**) CCSR for all assemblages; (**B**) CCSRs of each of four major assemblages occupying four major climate zones (biogeographic regions): Atlantic Arctic, Boreal Polar, Gulf Stream and Northwest Atlantic Shelves.

**Figure 2 biology-09-00042-f002:**
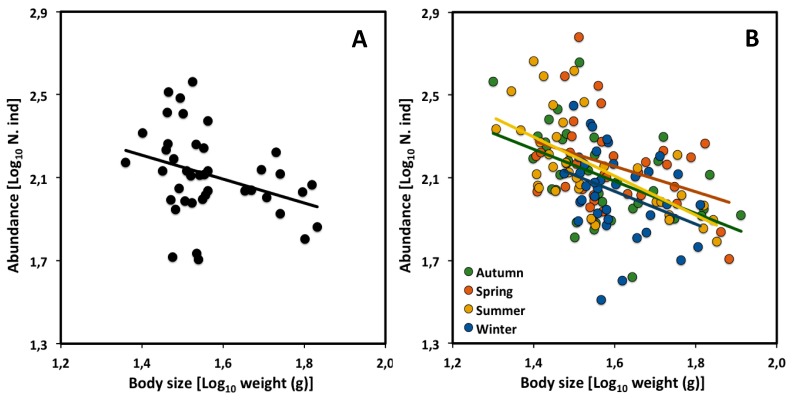
Scaling of density versus mean body size (live weight) of a desert rodent community in Portal Arizona through 41 years, from 1978 to 2018 (updated data from [34]): (**A**) CCSR for all assemblages; (**B**) CCSRs for each of four seasons (autumn, spring, summer, and winter) through 39 years (following the data cleaning described in Appendix A).

**Figure 3 biology-09-00042-f003:**
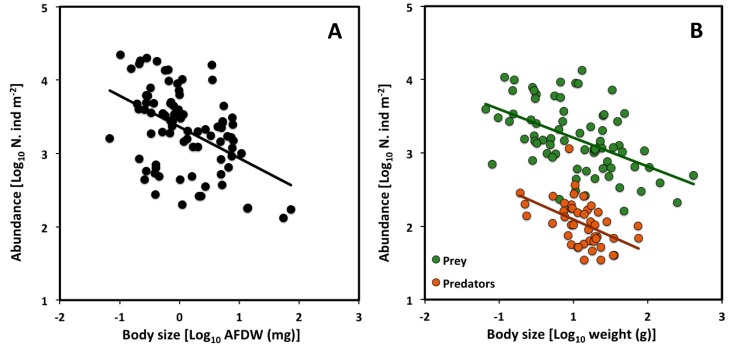
Scaling of population density versus body size (AFDW: ash free dry weight) across macroinvertebrate communities in Mediterranean and Black Sea lagoons (data from [29,30]): **A**) CCSR for all assemblages at various sampling sites; (**B**) CCSRs for prey and predators analyzed separately across sampling sites. Note that since the mean body size and population density were analyzed separately for prey and predator species across sampling sites, the number of points are greater and located at different positions in graph (**A**) vs. graph (**B**).

**Table 1 biology-09-00042-t001:** Results of the LSR analyses of log10 population density (cell/m^−2^) in relation to log10 body size (cell carbon) of phytoplankton communities in the Atlantic Ocean (data from [27]).

CCSR	Slope	95% CI	Intercept	*n*	*r^2^*	*p*
All assemblages	−0.68	−0.71 to −0.65	11.89	635	0.77	***
Atlantic Arctic	−0.39	−0.53 to −0.24	12.07	59	0.33	***
Boreal Polar	−0.44	−0.60 to −0.27	11.81	124	0.31	***
Gulf Stream	−0.67	−0.77 to −0.71	11.96	31	0.77	***
NW Atlantic Shelves	−0.74	−0.78 to −0.56	11.84	479	0.85	***

*** *p* < 0.001.

**Table 2 biology-09-00042-t002:** P value for slope and intercept comparison of the LSR analyses in Table 1. The differences among slopes were assessed by comparing 95% CI. When the slopes were not significantly different, the differences between elevations were estimated by ANCOVA (with body mass as a covariate).

Climate Zone	*p* Value for Slope ^a^				*p* Value for Intercept ^b^			
	AA	BP	GS	NWAS	AA	BP	GS	NWAS
Atlantic Arctic (AA)	-	ns	***	***	-	***	-	-
Boreal Polar (BP)	ns	-	***	***	***	-	-	-
Gulf Stream (GS)	***	***	-	ns	-	-	-	**
NW Atlantic Shelves (NWAS)	***	***	ns	-	-	-	**	-

^a^ Significance of slope differences; ^b^ Significance of intercept differences; ** *p* < 0.005; *** *p* < 0.001; ns—not significant; - not measurable.

**Table 3 biology-09-00042-t003:** Results of LSR analyses of log10 population abundance (number of individuals) in relation to log10 body size (mg) of a desert rodent community in Portal Arizona (updated data from [34]).

CCSR	Slope	95% CI	Intercept	*n*	*r^2^*	*p*
All assemblages	−0.55	−1.06 to −0.03	2.96	41	0.11	*
Autumn	−0.77	−1.24 to −0.31	3.32	38	0.25	**
Spring	−0.61	−1.16 to −0.07	3.14	38	0.13	*
Summer	−0.94	−1.28 to −0.22	3.62	38	0.42	***
Winter	−0.79	−1.47 to −0.25	3.30	38	0.12	*

* *p* < 0.05; ** *p* < 0.005; *** *p* < 0.001.

**Table 4 biology-09-00042-t004:** *p* values for slope and intercept comparisons of the LSR analyses in Table 3. The differences among slopes were assessed by comparing 95% CI. When the slopes were not significantly different, the differences between elevations were estimated by ANCOVA (with body mass as a covariate).

Seasons	*p* Value for Slope ^a^				*p* Value for Intercept ^1 b^			
	AU	SP	SU	WI	AU	SP	SU	WI
Autumn (AU)	-	ns	ns	ns	-	ns	ns	ns
Spring (SP)	ns	-	ns	ns	ns	-	ns	**
Summer (SU)	ns	ns	-	ns	ns	ns	-	ns
Winter (WI)	ns	ns	ns	-	ns	**	ns	-

^1^ Note that the calculated intercepts lie far outside the range of observed data points. Therefore, although the CCSR intercept during spring is lower than that for all other seasons (Table 3) because of a shallow CCSR scaling slope, within the range of observed data points, the scaling elevation is highest for spring (see Figure 2B); ^a^ Significance of slope differences; ^b^ Significance of intercept differences; ** *p* < 0.005; ns—not significant; - not measurable.

**Table 5 biology-09-00042-t005:** Results of the LSR analyses of log10 population density (number of individuals per m^2^) in relation to log10 body size (AFDW: ash free dry weight) of macroinvertebrate communities in Mediterranean and Black Sea lagoons (data from [29,30]).

CCSR	Slope	95% CI	Intercept	*n*	*r^2^*	*p*
All assemblages	−0.43	−0.60 to −0.25	3.36	85	0.22	***
Prey	−0.39	−0.53 to −0.24	3.40	75	0.25	***
Predators	−0.45	−0.78 to −0.24	2.32	45	0.23	***

*** *p* < 0.001.

**Table 6 biology-09-00042-t006:** *p* values for slope and intercept comparisons of the LSR analyses in Table 5. The differences among slopes were assessed by comparing 95% CI. When the slopes were not significantly different, the differences between elevations were estimated by ANCOVA (with body mass as a covariate).

Trophic Level	*p* Value for Slope ^a^		*p* Value for Intercept ^b^	
	Prey	Predators	Prey	Predators
Prey	-	ns	-	***
Predators	ns	-	***	-

^a^ Significance of slope differences; ^b^ Significance of intercept differences; *** *p* < 0.001; ns—not significant; - not measurable.

**Table 7 biology-09-00042-t007:** CCSR slopes for various community assemblages of species, including field (e.g., phytoplankton, macroinvertebrate, amphibian, fish, bird and rodent assemblages) and experimental studies (e.g., algae, bacteria and protozoa) reported in the literature. 95% confidence intervals and significant deviation of the slopes from the theoretical expected value of −3/4 are also shown.

Assemblages	N	Slope	95% CI	Deviation from −3/4	Reference
Phytoplankton	656	−0.78	−0.74 to −0.81^1^	=	[1]
Algae, bacteria & protozoa	20	−0.35	−0.01 to −0.71 ^1^	>	[27]
	20	0.36	0.00 to −0.72 ^2^	>	
	20	−1.15	−0.96 to −1.34 ^3^	<	
	20	−1.34	−1.01 to−1.67 ^4^	<	
	20	−1.05	−1.21 to −0.89 ^5^	<	
	20	−1.02	−1.16 to −0.88 ^6^	<	
Macroinvertebrates	158	−0.27	−0.41 to −0.13^1^	>	[28]
Macroinvertebrates	75	−0.35	−0.55 to −0.23 ^7^	>	[29]
	68	−0.35	−0.62 to −0.08 ^8^	>	
	65	−0.58	−0.78 to −0.37 ^9^	>	
	64	−0.44	−0.58 to 0.31 ^10^	>	
	45	−0.45	−0.67 to−0.24 ^11^	>	
Macroinvertebrates	32	−0.36	−0.67 to −0.06	>	[30]
Amphibians, fishes	18	−0.64	−1.00 to −0.28	=	[31]
& macroinvertebrates					
Winter land birds	285	−1.00	−1.43 to −0.57	<	[32]
Desert rodents	25	−0.57	−0.97 to −0.18	=	[33]

^1^ Week 1; ^2^ Week 2; ^3^ Week 3; ^4^ Week 4; ^5^ Week 5; ^6^ Week 6; ^7^ Deposit feeders; ^8^ Suspension feeders; ^9^ Shredders/Scrapers; ^10^ Gathering Collectors; ^11^ Predators.

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
