# Peer review of "A Perspective on Body Size and Abundance Relationships across Ecological Communities"

_biology, 2020, doi:10.3390/biology9030042_

Round 1

Reviewer 1 Report

The main message of this paper is: the cross-community scaling relationships do not have the same exponent, once we break down the community into different groups. The paper exemplifies this with some case studies. This is a worthwhile endeavour, and an interesting addition to the literature.

Major points

Lines 72 and 76: It is not obvious to me how the slopes and intercepts were estimated. I believe they were obtained using typical procedures for fitting linear regressions, such as the function “lm” in R. If this is the case, this is equivalent to the least square method (see the Ecological Detective, for example). If this is the case why do the authors report different values, for example, lines 72 and 76.

Lines 145-147 – I think what the authors mean here is that the total amount of energy is a function of the season, and that forces the intercept to be different. In terms of the power law, the slope is the same, no matter what the season is.

Why are figures 3a and 3b different? I assumed they would have exactly the same points.\

Lines 183-185 – I don’t think there is an obvious connection here with the second law of thermodynamics. In fact, none of the references 50-52 makes such connection. I would not mention the second law of thermodynamics here.

Lines 250-252 – This is probably the sentence of the paper I have more problems with:

“Many kinds of biological and ecological scaling relationships may not follow universal laws, but may depend on various biological and ecological contexts”.

Is it possible that we simply do not know some other universal laws that explain why and how the slopes vary? The next sentence reinforces this idea, but I’m not so pessimistic, and although I see value in checking the validity of proposed “laws”, I would not so easily disregard the existence of universal laws.

As far as I am concerned this is an appeal to a simple collection of facts, without an attempt to unify them. I have to confess that this type of comments always make me rather apprehensive.

Minor points

Line 36 – missing parenthesis at the very end “)”.

Lines 46-48 – I con’t understand the following sentence: “The ‘metabolic theory of ecology’ (MTE) also predicts a -3/4 caling exponent, which is the inverse of the scaling exponent for metabolic rate, according to the 3/4-power law”.

Lines 52-54 – “A less steep slope may occur if large species acquire more energy than smaller species. In contrast, a steeper slope may occur if small species acquire more energy than larger species”. As far as I understand this sentence is correct, but to understand it the reader has to keep in mind the last sentence of the previous paragraph:

“This prediction assumes that each species population uses approximately the same amount of energy (calculated as energetic demand per individual times the number of individuals in a population), a concept called the Energetic Equivalence Rule”

Maybe this paragraph becomes easier to read it there is a connection between the two sentences (assuming that I interpret it correctly).

Line 67 – What does “unitary laws” mean?

Line 253 - What does “physical unitary explanations” mean?

Author Response

Reviewer 1

Comments and Suggestions for Authors

The main message of this paper is: the cross-community scaling relationships do not have the same exponent, once we break down the community into different groups. The paper exemplifies this with some case studies. This is a worthwhile endeavour, and an interesting addition to the literature.

Thank you for your encouragement. 

Major points

Point 1: Lines 72 and 76: It is not obvious to me how the slopes and intercepts were estimated. I believe they were obtained using typical procedures for fitting linear regressions, such as the function “lm” in R. If this is the case, this is equivalent to the least square method (see the Ecological Detective, for example). If this is the case why do the authors report different values, for example, lines 72 and 76.

Response 1: We indicate in the manuscript that in the original analysis by Li et al. (2001), reduced major axis analysis was used (L 73), whereas we used least squares regression (LSR) to determine the slopes and intercepts (L 75-77, 81).

Point 2: Lines 145-147 – I think what the authors mean here is that the total amount of energy is a function of the season, and that forces the intercept to be different. In terms of the power law, the slope is the same, no matter what the season is.

Response 2: Yes, this is what we mean. We have attempted to clarify this point.

Point 3: Why are figures 3a and 3b different? I assumed they would have exactly the same points.

Response 3: As indicated in the Fig. 3 legend, graph A provides the scaling analysis for all samples including prey and predator species together, whereas graph B provides separate scaling analyses for prey and predators across the sampling sites. Since the mean body size and population density were analyzed separately for prey and predator species across the sampling sites, the number of points are greater and located at different positions in Fig. 3A vs. Fig. 3B. We have clarified this point in the text and Fig. 3 legend.

Point 4: Lines 183-185 – I don’t think there is an obvious connection here with the second law of thermodynamics. In fact, none of the references 50-52 makes such connection. I would not mention the second law of thermodynamics here.

Response 4: We respectfully disagree. The loss of energy as heat as energy flows through food chains is classically explained in terms of the second law of thermodynamics. We made a mistake and should have cited reference 49 (the classic paper of Lindemann 1942) in this context. Although Lindemann (1942) did not explicitly use the phrase “second law of thermodynamics”, his concept that heat energy is dissipated as energy is transformed from one trophic level to another is based on this law. This concept is now widely accepted by ecologists. We have also added a citation to Odum (1968), who explicitly mentions the role of the 2nd law in ecosystem energy flow.

Point 5: Lines 250-252 – This is probably the sentence of the paper I have more problems with:

“Many kinds of biological and ecological scaling relationships may not follow universal laws, but may depend on various biological and ecological contexts”.

Is it possible that we simply do not know some other universal laws that explain why and how the slopes vary? The next sentence reinforces this idea, but I’m not so pessimistic, and although I see value in checking the validity of proposed “laws”, I would not so easily disregard the existence of universal laws.

As far as I am concerned this is an appeal to a simple collection of facts, without an attempt to unify them. I have to confess that this type of comments always make me rather apprehensive.

Point 5: We have reworded this section to make it clearer. We do not reject universal laws, but rather claim that various biological and ecological factors may cause deviation from them.

Minor points

Point 1: Line 36 – missing parenthesis at the very end “)”.

Response 1: We have added the parenthesis.

Point 2: Lines 46-48 – I can’t understand the following sentence: “The ‘metabolic theory of ecology’ (MTE) also predicts a -3/4 scaling exponent, which is the inverse of the scaling exponent for metabolic rate, according to the 3/4-power law”.

Response 2: We have reworded this sentence and the following one to improve clarity.

Point 3: Lines 52-54 – “A less steep slope may occur if large species acquire more energy than smaller species. In contrast, a steeper slope may occur if small species acquire more energy than larger species”. As far as I understand this sentence is correct, but to understand it the reader has to keep in mind the last sentence of the previous paragraph:

“This prediction assumes that each species population uses approximately the same amount of energy (calculated as energetic demand per individual times the number of individuals in a population), a concept called the Energetic Equivalence Rule”

Maybe this paragraph becomes easier to read it there is a connection between the two sentences (assuming that I interpret it correctly).

Response 3: Yes, there is a connection, which is why the statements above are stated in succession. We hope that we have made this clearer now.

Point 4: Line 67 – What does “unitary laws” mean?

Response 4: We changed “unitary” to “universal”.

Point 5: Line 253 - What does “physical unitary explanations” mean?

Response 5: Reworded for clarification.

Reviewer 2 Report

This is a highly-important and informative contribution that addresses an important problem of the ecology. With a series of examples, the authors demonstrate the complexity of the considered problem and some ideas for its further examination. The paper is well-written, and it bears a representative list of references. The structuring of the paper is non-standard, but this is a commentary, and, thus, the existing structure is appropriate.

I strongly recommend this paper for acceptance after MINOR improvements. Two my suggestions are as follows.

Why have you preferred THESE examples? The relevant explanation in the beginning of your paper will be very helpful to the readers. Please, add some else "fresh" literature sources. I encourage the authors to think on addition of a summary table comparing the "classical" views of body size-abundance relationship and with the conclusions made by the authors themselves.

I wish to add some outcomes of this study are highly-important in palaeoecology where the problem of body size is actively discussed. I'd prefer seeing some relevant notes in this manuscript, but I do not insist on doing this as the authors are not palaeontologists.

Author Response

Reviewer 2

Comments and Suggestions for Authors

This is a highly-important and informative contribution that addresses an important problem of the ecology. With a series of examples, the authors demonstrate the complexity of the considered problem and some ideas for its further examination. The paper is well-written, and it bears a representative list of references. The structuring of the paper is non-standard, but this is a commentary, and, thus, the existing structure is appropriate.

Thank you for your encouragement.

Point 1: I strongly recommend this paper for acceptance after MINOR improvements. Two of my suggestions are as follows.

Why have you preferred THESE examples? The relevant explanation at the beginning of your paper will be very helpful to the readers. Please, add some else "fresh" literature sources. I encourage the authors to think on the addition of a summary table comparing the "classical" views of body size-abundance relationships and with the conclusions made by the authors themselves.

Response 1: This is a nice suggestion, but our focus is on the scaling of abundance at the community level, not at the population level. Most of the classical theory applies to the population level. The analysis of CCSRs has occurred only recently, and we would like to provide recommendations for how this research should be conducted in the future.   To do so, we make our recommendations in the context of three examples involving the effects of different ecological factors. We hope that we have clarified this point in the Introduction.

Point 2: I wish to add some outcomes of this study are highly-important in palaeoecology where the problem of body size is actively discussed. I'd prefer seeing some relevant notes in this manuscript, but I do not insist on doing this as the authors are not paleontologists.

Response 1: Unfortunately, we do not know of any CCSRs based on fossil data, so we cannot make any comments about this. Note that it is very difficult to obtain reliable data on the abundance of organisms in fossil communities.